# Exploring Biomolecular Self-Assembly with Far-Infrared Radiation

**DOI:** 10.3390/biom12091326

**Published:** 2022-09-19

**Authors:** Takayasu Kawasaki, Yuusuke Yamaguchi, Hideaki Kitahara, Akinori Irizawa, Masahiko Tani

**Affiliations:** 1Accelerator Laboratory, High Energy Accelerator Research Organization, 1-1 Oho, Tsukuba 305-0801, Ibaraki, Japan; 2Research Center for Development of Far-Infrared Region, University of Fukui, 3-9-1 Bunkyo, Fukui 910-8507, Fukui, Japan; 3SR Center, Research Organization of Science and Technology, Ritsumeikan University, 1-1-1 Nojihigashi, Kusatsu 525-8577, Shiga, Japan

**Keywords:** terahertz, far-infrared radiation, amyloid, cellulose, free-electron laser, gyrotron, self-assembly

## Abstract

Physical engineering technology using far-infrared radiation has been gathering attention in chemical, biological, and material research fields. In particular, the high-power radiation at the terahertz region can give remarkable effects on biological materials distinct from a simple thermal treatment. Self-assembly of biological molecules such as amyloid proteins and cellulose fiber plays various roles in medical and biomaterials fields. A common characteristic of those biomolecular aggregates is a sheet-like fibrous structure that is rigid and insoluble in water, and it is often hard to manipulate the stacking conformation without heating, organic solvents, or chemical reagents. We discovered that those fibrous formats can be conformationally regulated by means of intense far-infrared radiations from a free-electron laser and gyrotron. In this review, we would like to show the latest and the past studies on the effects of far-infrared radiation on the fibrous biomaterials and to suggest the potential use of the far-infrared radiation for regulation of the biomolecular self-assembly.

## 1. Introduction

In these days, far-infrared radiation has been frequently employed in chemical and biomedical research fields. For example, microwave heating can be applicable for solid-phase peptide synthesis and metal-based nanoparticle preparations [1,2]. In addition, far-infrared lasers are often employed for biomedical research using cancer cell lines to develop phototherapies and photo-diagnostics [3,4,5]. The high permeability to the biological tissue is also effective for in vivo bioimaging [6]. Nonetheless, the target substances are often heated by the radiation, and while the far-infrared radiation can cause remarkable structural changes of target biomolecules, whether the radiation effect is thermal or non-thermal remains unclear. In Figure 1, frequencies and wavelengths of lights (upper) and parameters of the oscillation systems employed in our study are shown. The wavelengths in the terahertz region usually range from 30 to 3000 μm and are applied to various studies such as spectroscopy [7], radiation [8], and spectral imaging [9]. We focused on a free-electron laser (FEL) and a submillimeter wave from a gyrotron and recently discovered that the high-power far-infrared radiation can regulate self-association of proteins at different far-infrared wavelengths [10,11,12]. The terahertz FEL (so called THz-FEL) has a double pulse structure that is composed of micro- and macro-pulses, in which the duration of the former is 10–20 ps and that of the latter is 4 μs [13]. The oscillation wavelength covers from 30 to 300 μm, and the irradiation power is given as avg. 5 mJ per macropulse (see Appendix A for the oscillation system). The submillimeter wave from the gyrotron is a single pulse of 1–2 ms half width having 10 W power [14]. The gyrotron oscillation system (Appendix A) nowadays acts as a strong radiation source for nuclear magnetic resonance (NMR) spectroscopy with the dynamic nuclear polarization (DNP) technique [15,16,17]. In addition, hyperthermia treatment using far-infrared radiation is expected to be a candidate for the therapeutic management of cancer [18,19].

## 2. Dissociation of Amyloid Fibril by THz-FEL Irradiation

Amyloid fibril is known to be involved in various biological phenomena such as the onset of serious amyloidosis [20,21], biofilm formation [22,23], biosynthesis of pigment melanin [24], protection of eggshells [25], supramolecular assembly in the body structures [26], and gene expressions [27]. In addition, amyloid fibril can be utilized as functional biomaterials such as rigid scaffolds for cell cultivation and tissue engineering [28,29], artificial capsules and hydrogels for drug delivery systems [30,31], and functional nanofilms for microorganism adhesion [32]. Hen-egg white lysozyme is a glycoside hydrolase composed of 129 amino acids and forms amyloid-like fibril under acidic conditions [33,34]. In the far-infrared spectra from 130 to 250 cm^−1^ (Figure 2a), the protein exhibits a strong absorption peak at about 170–190 cm^−1^ (52.6–58.8 μm) and weak absorption peaks at 130–150 cm^−1^ (66.7–76.9 μm). The THz-FEL was tuned to 56 or 70 μm and impinged onto the lysozyme fibril for 10 min at room temperature. There are two bands (1620 and 1650 cm^−1^) at the amide I region before irradiation (Figure 2b, black line), and the irradiation at 56 μm showed a decrease in the peak intensity at a lower wavenumber and an increase of that at a higher wavenumber (red line). Since the amide I band at the lower wavenumber corresponds to β-sheet and the latter band influences α-helix or non-fibrous conformations [35,36], the spectral change by the irradiation indicates the decrease of the β-sheet-rich conformation. The conformational analysis based on the intensity at the amide I band (Figure 2c) proved that the proportion of β-sheet decreased from 45% before irradiation (black bar) to 20% after irradiation (red bar) and that of α-helix increased from 5% (black bar) to 30% (red bar) from the irradiation. β-Turn and other conformations are only slightly changed by the irradiation. On the other hand, the irradiation at 70 μm (green line and bar) also showed the similar effect, but the degree of conformational change is not remarkable. Congo red staining (Figure 2d, upper) clearly showed the irradiation spot (white solid circle) distinct from the safe fibril state (dotted circle), and in the SEM images (below), assemblies consisting of many strings (several nanometers in width and several micrometers in length) were destroyed by the irradiation at 56 μm. 

Previously, we reported that the THz-FEL can dissociate an amyloid fibril of calcitonin DFNKF peptide [10], and the peptide fibril was dissociated similarly with the lysozyme fibril. Together with this prior study, it can be implied that several kinds of amyloid fibrils can be decomposed by the terahertz laser irradiation. Although the detailed mechanism is not clear, it can be considered that the dissociation process may be similar to the phenomenon where a solid aggregate is momentarily unraveled in boiling water: the extreme thermal energy is absorbed into the fibril structure at the collective vibrational modes. Regarding normal proteins, the FEL irradiation little damaged the whole structure and enzymatic activity of native lysozyme [34]. Therefore, the THz-FEL can be tested as a therapeutic strategy for amyloidosis in surgical medicine in future [37]. In addition, the terahertz laser can also be tested to regulate the growth of microorganisms because the biofilm formation is closely related with amyloids [38]. 

## 3. Promotion of Amyloid Fibrillation by the Submillimeter Wave from Gyrotron

Next, an effect of the submillimeter wave on the protein fibril is shown. The submillimeter wave was tuned to 720 μm and oscillated by a 420 GHz gyrotron (Appendix A). Prior to the irradiation experiments, we confirmed that the transmittance of the submillimeter wave against the Eppendorf tube was more than 80% at the 0–2.0 THz region (Appendix A). The temperature increase on the surface of the sample in the tube was only around 5 K compared to the non-irradiation area during the irradiation (Figure 3a). In the mid-infrared spectra (Figure 3b), the peak intensity at around 1620 cm^−1^ was apparently increased after the irradiation (red) compared to that of before irradiation (black), and the conformational analysis (Figure 3c) indicated that β-sheet was increased and α-helix was decreased after the irradiation. Congo red staining (Figure 3d, upper) showed that fibrils were apparently increased and SEM observation (bottom) revealed that the fibril structure changed into solid aggregates by the submillimeter wave. The result by X-ray scattering analysis is shown in Figure 3e. The inclination of the scattering curve from 3 nm^−1^ to 9 nm^−1^ was larger after the irradiation (red) than that before irradiation (black), which means that the shape of the aggregate was changed into the thick lamellar type [39]. A scattering peak at around 3.8 nm^−1^ corresponds to 1.65 nm of the fibril layer, and this value is quite larger compared to the typical size (0.9–1.0 nm) of amyloid fibril [40]. In addition, a tiny peak at 10.6 nm^−1^ indicates that the protein forms a cross-β-sheet conformation and the interval distance between β-sheet chains is 0.59 nm. This value is slightly larger than the typical size (0.4–0.5 nm) [41]. 

We demonstrated that the submillimeter wave can promote the fibril formation of many kinds of peptides and proteins (GNNQQNY, Aβ_1-40_, SAA, DFNKF, and lysozyme) [11]. In every case, β-sheet was dominated and the conformation was more aggregated than the pre-irradiation state. The reformed aggregate seems to be shaped larger and more rigid than the pre-irradiation state. Therefore, it can be implied that the fibrous characteristics such as rigidness and regularity can be altered by the irradiation. As for the mechanism, it can be considered that the submillimeter wave can activate the intermolecular motions and induce conformational changes in the fibrous structure. Such molecular shaking can trigger the spontaneous association of the monomer chains to produce the stacking conformation even without any external heating [42].

## 4. Dissociation and Re-Association of Cellulose Fiber with Far-Infrared Radiation

We explored the applicability of the far-infrared radiation to the other fibrous biomaterials. Cellulose fiber can be developed for cosmetic additives, anti-bacterial sheets, and porous materials in healthcare and pharmaceuticals fields [43,44,45]. In addition, cellulose fibers are applied for components of electronic devices and auto parts in mechanical industries [46,47]. In the mid-infrared spectrum (Figure 4a), a strong band at about 1050 cm^−1^ and middle peak around 1300 cm^−1^ were observed (black). The former band corresponds to the stretching vibrational mode of the glycoside bond (νC-O), and the latter peak can be assigned to bending vibration of H-C-O, respectively [48]. When the cellulose was irradiated by the THz-FEL tuned to 80 μm, the former peak was decreased and the latter peak was increased (blue). On the other hand, the former peak was largely increased after irradiation by the submillimeter wave at 720 μm (red). Interestingly, when the cellulose fiber was irradiated by the submillimeter wave after the THz-FEL (green), the whole spectral pattern was almost the same as that of the original cellulose (black). Therefore, it can be considered that the fiber was dissociated to the monomeric form by the THz-FEL, and the monomers were re-aggregated by the submillimeter wave radiation. In the morphological images (Figure 4b), solid fibers that are several hundred micrometers in length before irradiation (upper left, yellow circle) were destroyed into a number of small particles after THz-FEL irradiation (below). On the contrary, irradiation by submillimeter wave assembled the fibers more than the original cellulose (upper right). We have ever used a mid-infrared FEL to dissociate the cellulose aggregate for the purpose of obtaining the monomeric sugars for the biorefinery application [49]. The mid-infrared FEL is an accelerator-based intense pulse laser at mid-infrared wavelengths (5–10 μm). The time structure is composed of micro-pulse and macro-pulse, and each duration is 1–2 ps and 2 μs, respectively. The oscillation energy is about 5–10 mJ per a macro-pulse. When the cellulose fiber was irradiated by the mid-infrared FEL at 9.1 μm (νC-O), the absorption bands at 1080–1090 cm^−1^ were largely reduced (Figure 4c, blue enclosure). This indicates that the glycoside bonds were dissociated and low-molecular weight oligosaccharides were released. On the contrary, the glycoside band around 1080 cm^−1^ survived after the THz-FEL irradiation (Figure 4a). Therefore, it can be considered that the cleavage of the glycoside bonds is unlikely and the non-covalent bonds such as hydrogen bonds can be affected in the case of the far-infrared radiation.

## 5. Future Aspect of the Use of Far-Infrared Radiation in Biological and Material Fields

The above results suggest that the fibrous conformations of biomacromolecules were dissociated by the terahertz laser and associated by the submillimeter wave. This study implies that regulation of self-assembly of biomaterials can be performed in one test tube by using far-infrared radiations at different wavelengths. Nonetheless, more consideration regarding the difference in the irradiation effects between the THz-FEL and the submillimeter wave from the gyrotron should be needed because not only wavelength but also time structure is varied. The peak power of a macro-pulse of THz-FEL is quite larger than the case of the gyrotron due to the shorter pulse duration (Figure 1). It can also be estimated that one of factors for regulating the aggregation process of biomolecules is the radiation power. Although the study on this reaction mechanism should be a next subject, the fiber biomaterials can be recycled without any organic solvents or external heating by using both the free-electron lasers and gyrotron continuously, which inspires the use of electromagnetic waves at the terahertz region to a sustainable engineering system of solid textile biomaterials. In addition, it can be expected that several types of structural proteins such as silk fibroin [50,51], keratin fiber [52,53], laminin [54,55], and elastin [56,57] can also be targeted by the far-infrared radiation. Those proteins form highly regularized aggregates, although the molecular sizes are various. It is very interesting how the far-infrared radiation can activate those vibrational states and alter the cell functions related with the biomolecular self-associations.

## 6. Conclusions

We demonstrated that fibrous biomolecules can be dissociated and re-associated by using two kinds of far-infrared radiation, THz-FEL and submillimeter wave from a gyrotron. THz-FEL can dissociate the stacking conformations of amyloid fibrils with a decrease in β-sheet and increase in α-helix, and the submillimeter wave can promote the fibrillations reversely. The cellulose fiber was dissociated by the THz-FEL and re-aggregated by the submillimeter wave in a similar manner. Those reactions can be performed at ambient temperature without any external heating. The combinatorial use of these far-infrared radiations is expected to regulate the fibrillar biomolecules and contribute to biomolecular engineering in biology and biomaterial fields.

## Figures and Tables

**Figure 1 biomolecules-12-01326-f001:**
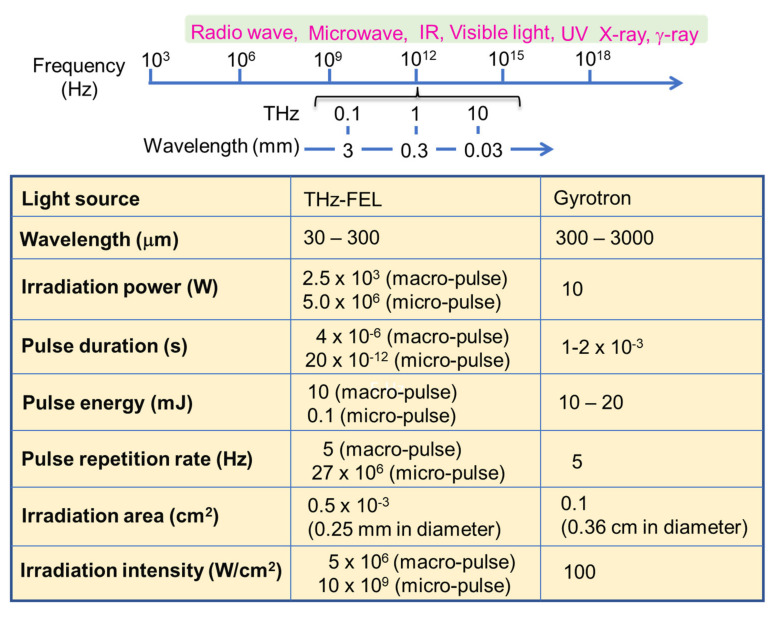
Frequency region of electromagnetic waves (upper) and far-infrared radiation parameters used in this study.

**Figure 2 biomolecules-12-01326-f002:**
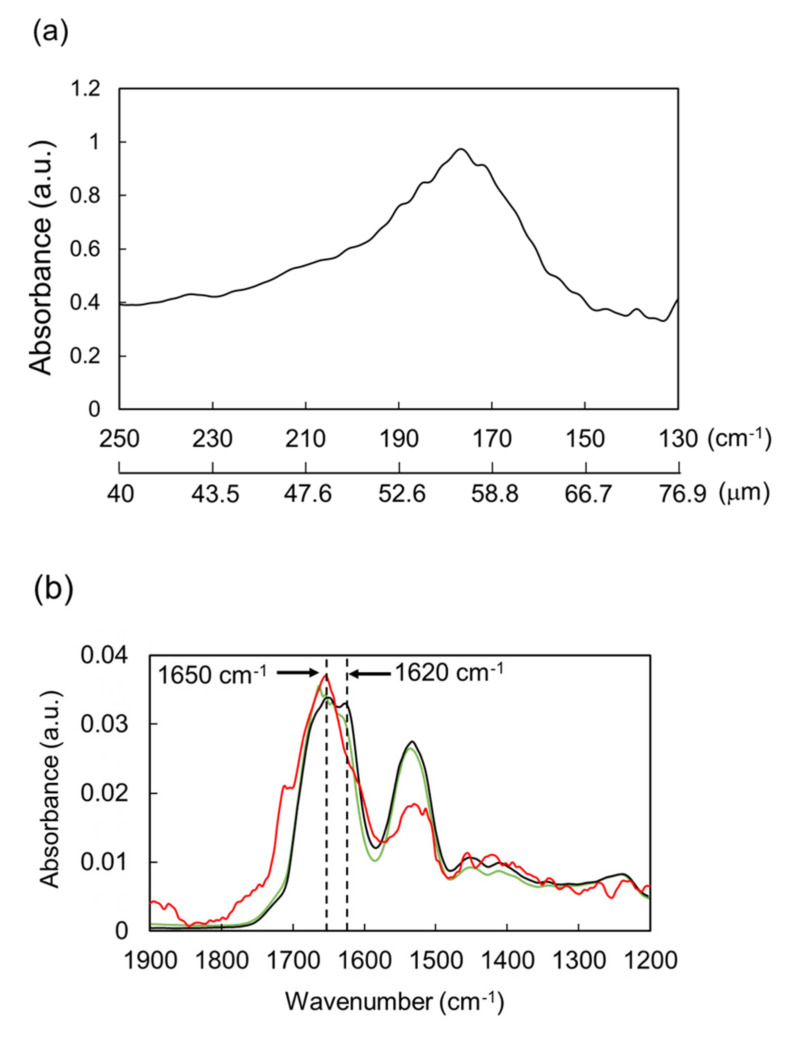
Irradiation Effect of THz-FEL on lysozyme fibril. (**a**) Far-infrared spectrum. (**b**) Mid-infrared spectra. Black: non-irradiation; red: irradiation at 56 μm; green: irradiation at 70 μm. (**c**) Protein secondary conformations before and after irradiation. The color category is the same as (**b**). (**d**) Congo red staining (upper) and SEM observation (below). Irradiation area is shown as a white solid circle and non-irradiation area is indicated by a white dotted circle. White scale bar: 500 μm; black scale bar: 200 nm.

**Figure 3 biomolecules-12-01326-f003:**
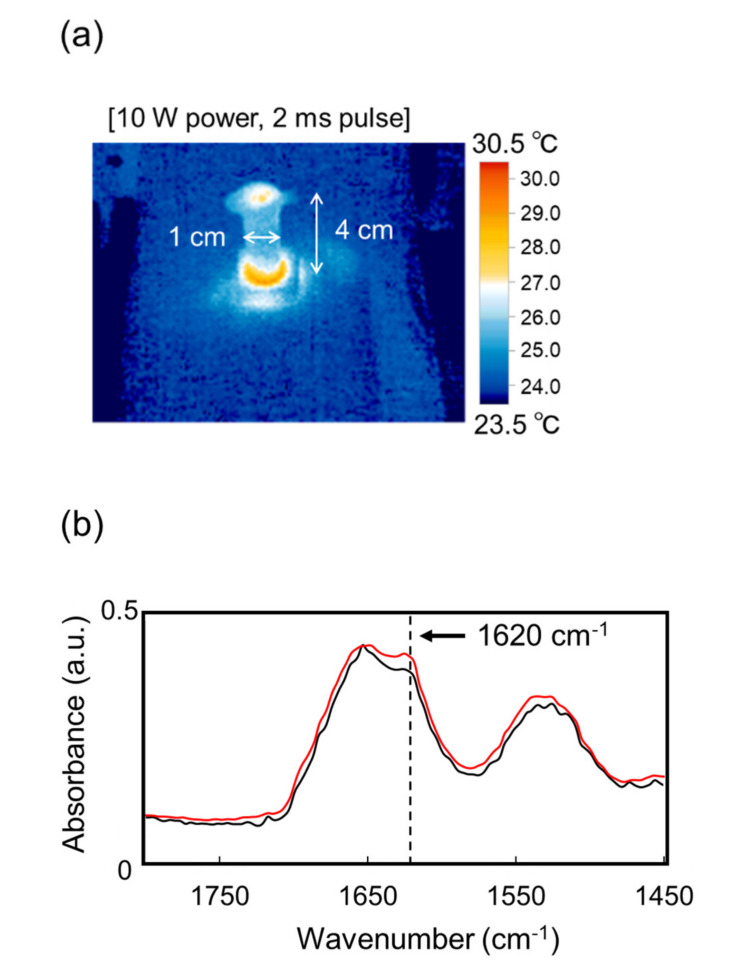
Irradiation Effect of submillimeter wave on lysozyme fibril. (**a**) Thermography camera observation. (**b**) Mid-infrared spectra before (black) and after (red) irradiation. (**c**) Proportions of protein secondary conformations. The color category is the same as (**b**). (**d**) Congo red staining (upper) and SEM observation (bottom) before (−) and after (+) irradiation. White bar: 500 μm; black bar: 1 μm. (**e**) SAXS spectra before (black) and after (red) irradiation: d value equals 2 π q^−1^.

**Figure 4 biomolecules-12-01326-f004:**
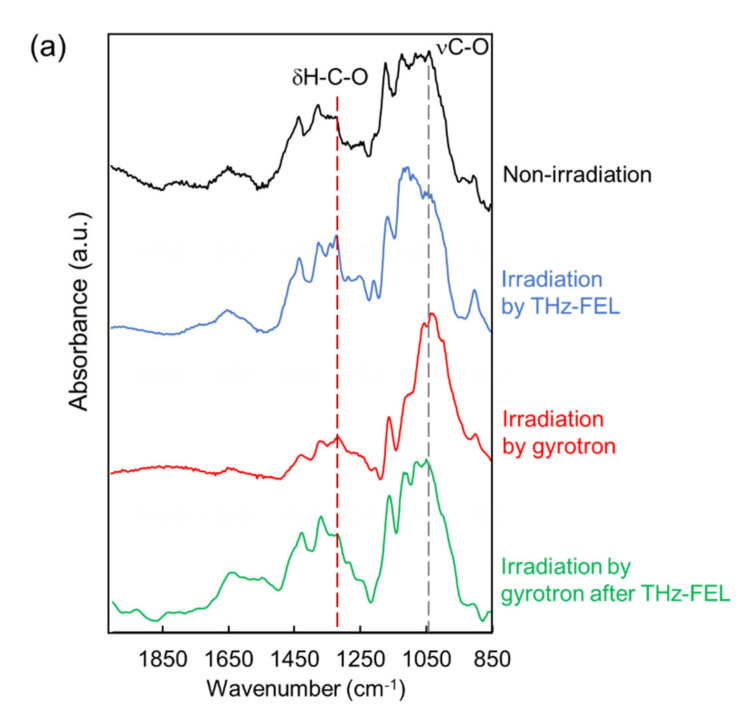
Irradiation effects of THz-FEL and submillimeter wave on cellulose fiber. (**a**) Infrared spectra. Black: non-irradiation; blue: irradiation with THz-FEL; red: irradiation with submillimeter wave; green: irradiation with submillimeter wave behind THz-FEL. (**b**) SEM observation. Upper left: non-irradiation; upper right: irradiation by submillimeter wave; below: irradiation by THz-FEL. Yellow bar: 100 μm. (**c**) Infrared spectra before (black) and after (red) irradiation by mid-infrared FEL at 9.1 μm.

## Data Availability

A part of the data is presented as Appendix A, and other experimental data are available from the corresponding author (T.K.).

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
