# Peer review of "Exploring Biomolecular Self-Assembly with Far-Infrared Radiation"

_biomolecules, 2022, doi:10.3390/biom12091326_

Round 1

Reviewer 1 Report

In this paper the authors report on recent and past studies on the irradiation effects of the terahertz waves on fibrous biomaterials and to suggest the potential use of far-infrared radiation for regulating  biomolecular self-assembly processes. The paper is well written and clearly presents the experimental method and the results obtained.

Two different types of THz radiation sources are used in this study: a gyrotron and a FEL, which have intrinsically different characteristics in terms of mode operation and output power. These peculiar different characteristics can be exploited to investigate and discriminate between the various biomolecular processes, in particular regarding self-assembly.

However, the operating parameters have to be better specified in a way that facilitate the comparison of the irradiation conditions. To this purpose I suggest that the table in Fig.1 should be improved by showing the following columns (parameters):

Light source                        (as in the current table)

Wavelength (micrometers)

Irradiation power (W)                   (in the current table the FEL macropulse energy appears in the table)

Pulse duration (s)                           (as in the current table)

Pulse energy (mJ)

Pulse repetition rate (Hz)

Irradiation area (cm2)

Irradiation intensity (W/cm2)

For comparison purposes, it would also be useful to show the gyrotron and FEL irradiation setups and sample geometry in figs. S1 and S2 respectively.

Minor typo in the Supporting Information - 2. Sample preparation:

“stain less steel” should read “stainless steel”

The paper is recommended for publication, once the above minor revision is done.

Author Response

Dear Reviewer 1

Thank you very much for reviewing our paper. We made responses to all of your comments as described below and revised the manuscript. We would appreciate it if you could review the responses and the revised paper.

Comment 1: However, the operating parameters have to be better specified in a way that facilitate the comparison of the irradiation conditions. To this purpose I suggest that the table in Fig.1 should be improved by showing the following columns (parameters):

Light source (as in the current table),

Wavelength (micrometers)

Irradiation power (W)(in the current table the FEL macropulse energy appears in the table)

Pulse duration (s)(as in the current table)

Pulse energy (mJ)

Pulse repetition rate (Hz)

Irradiation area (cm2)

Irradiation intensity (W/cm2)

Answer 1: Thank you so much for the valuable comment. As you suggested, we added several parameters for better comparison of the two-kinds of light sources in the revised Fig. 1 (line 52-57). We would appreciate it if you could consider this revision.

Comment 2: For comparison purposes, it would also be useful to show the gyrotron and FEL irradiation setups and sample geometry in figs. S1 and S2 respectively.

Answer 2: Thank you so much for the comment. As you suggested, we added the irradiation setups and sample geometry in the Figs. S1 and S2 in the revised version. We would appreciate it if you could find the revised figures and consider this revision.

Comment 3: Minor typo in the Supporting Information - 2. Sample preparation:

“stain less steel” should read “stainless steel”

Answer 3: Thank you for the comment. As you indicated, we revised the term from “stain less” to “stainless” in section 2 of Supporting Information.

That’s all.

Sincerely yours

Takayasu Kawasaki, Ph.D.

Accelerator Laboratory, High Energy Accelerator Research Organization, 1-1 Oho, Tsukuba, Ibaraki 305-0801, Japan.

Phone: +81-29-864-5200-2014, Fax: +81-29-864-3182

E-mail: takayasu.kawasaki@kek.jp

Reviewer 2 Report

The authors review their works on the irradiation effect of intense far-infrared radiation on biomolecules, particularly focusing on their fibrous structure. They have discovered that the structural change depends on the wavelength of the radiation. They suggest its potential use for controlling the biomolecular self-assembly.

The results are interesting. I think this paper should be accepted for publication. However, before the publication, the authors should consider following issues and improve the manuscript.

1.       The authors use various terms for the radiation as follows; far infrared radiation, terahertz wave, terahertz radiation, far infrared rays, far infrared light, submillimeter wave, THz FEL, terahertz laser... If the authors want to use these terms separately, they should state the definitions, otherwise they should unify them as much as possible.

2.       Figure 3(a): The authors should add information on the size.

3.       Line 180, “mid-infrared FEL” suddenly appears without description. It is better to describe it as THz-FEL or gyrotron.

4.       The authors seem emphasizing that the difference of the wavelength of the far infrared rays makes different effect on the biomolecules. However, the time structures of the rays are also different. I suggest authors to give a comment on this.

Author Response

Dear Reviewer2,

Thank you very much for reviewing our paper. We made responses to all of your comments as described below and revised the manuscript. We would appreciate it if you could review the responses and the revised paper.

Comment 1: The authors use various terms for the radiation as follows; far infrared radiation, terahertz wave, terahertz radiation, far infrared rays, far infrared light, submillimeter wave, THz FEL, terahertz laser... If the authors want to use these terms separately, they should state the definitions, otherwise they should unify them as much as possible.

Answer 1: Thank you so much for the valuable suggestion. As you suggested, it seems that the present description would confuse the readers. Basically, far-infrared radiations include both of terahertz lasers and submillimeter waves, and the terahertz region is a part of the far-infrared region. Therefore, we changed the term “terahertz radiation” into “radiation at terahertz region” (line 13-14) and “far-infrared radiation” (line 20, 21, 41, 50, 160-161, 217, 219, 224) for the unification. In addition, a keyword “terahertz waves” was changed into “terahertz” (line 24). In the case of gyrotron, the instrument can cover the wide-range far-infrared wavelengths, and we used a specific gyrotron at a submillimeter wave (720 um) in this study. Therefore, the term “submillimeter wave” should be kept to distinguishing from the THz-FEL and we changed the term “terahertz” into “submillimeter” in the description about gyrotron (section 3, line 115). In addition, we changed “rays” into “radiations” (line 28, 205).

We appreciate it if you could consider the above revision.

Comment 2: Figure 3(a): The authors should add information on the size.

Answer 2: Thank you for the comment. We added both sizes of diameter and height of the sample tube as shown in the revised Fig. 3(a).

Comment 3: Line 180, “mid-infrared FEL” suddenly appears without description. It is better to describe it as THz-FEL or gyrotron.

Answer 3: Thank you so much for the suggestion. We added a description about the mid-IR FEL as follows (line 179-184). “We have ever used a mid-infrared FEL to dissociate the cellulose aggregate for the purpose of obtaining the monomeric sugars for the biorefinery application [49]. The mid-infrared FEL is an accelerator-based intense pulse laser at mid-infrared wavelengths (5-10 um). The time structure is composed of micro-pulse and macro-pulse, and each duration is 1-2 ps and 2 us, respectively. The oscillation energy is about 5-10 mJ per a macro-pulse.” We would appreciate it if you considered the above additional phrase.

Comment 4: The authors seem emphasizing that the difference of the wavelength of the far infrared rays makes different effect on the biomolecules. However, the time structures of the rays are also different. I suggest authors to give a comment on this.

Answer 4: Thank you so much for the valuable suggestion. We added a comment about the different effects of two-kinds of far-infrared radiations to line 206-213 as follows: “Nonetheless, more consideration regarding the difference in the irradiation effects between the THz-FEL and the submillimeter wave from gyrotron should be needed because not only wavelength but also time structure is varied. The peak power of a macro-pulse of THz-FEL is quite larger than the case of gyrotron due to the shorter pulse duration (Fig. 1). It can also be estimated that one of factors for regulating the aggregation process of biomolecules is the radiation power. Although the study on this reaction mechanism should be a next subject, the fiber biomaterials …..”

We would appreciate it if you could consider the above additional description.

That’s all.

Sincerely yours

Takayasu Kawasaki, Ph.D.

Accelerator Laboratory, High Energy Accelerator Research Organization, 1-1 Oho, Tsukuba, Ibaraki 305-0801, Japan.

Phone: +81-29-864-5200-2014, Fax: +81-29-864-3182

E-mail: takayasu.kawasaki@kek.jp
